# A Study on the Static Magnetic and Electromagnetic Properties of Silica-Coated Carbonyl Iron Powder after Heat Treatment for Improving Thermal Stability

**DOI:** 10.3390/ma15072499

**Published:** 2022-03-28

**Authors:** Xu Yan, Xinyuan Mu, Qinsheng Zhang, Zhanwei Ma, Chengli Song, Bin Hu

**Affiliations:** State Key Laboratory for Oxo Synthesis and Selective Oxidation, Lanzhou Institute of Chemical Physics, Chinese Academy of Sciences, Lanzhou 730000, China; yanxu@licp.cas.cn (X.Y.); muxinyuan@licp.cas.cn (X.M.); zhangqinsheng@licp.cas.cn (Q.Z.); mazhanwei@licp.cas.cn (Z.M.)

**Keywords:** microwave absorption, thermal stability, silica coating, carbonyl iron powders

## Abstract

In order to study the thermal stability of coated carbonyl iron powder (CIP) and its influence on magnetic properties, carbonyl iron powder was coated with a silica layer and then annealed in an air atmosphere at elevated temperatures. Transmission electron microscopy (TEM) analysis and Fourier transform infrared spectroscopy confirmed the existence of a silicon dioxide layer with a thickness of approximately 80~100 nm. Compared with uncoated CIP, the silicon-coated CIP still maintained a higher absorption performance after annealing, and the calculated impedance matching value Z only slightly decreased. It is worth noting that when the annealing temperature reached 300 °C, coercivity (*H_c_*) increased, and the real and imaginary parts of the permeability decreased, which means that the silicon dioxide layer began to lose its effectiveness. On the contrary, the significant decrease in microwave absorption ability and impedance matching value Z of uncoated CIP after annealing were mainly because the newly formed oxide on the interface became the active polarization center, leading to an abnormal increase in permittivity. In terms of the incremental mass ratio after annealing, 2% was a tipping point for permeability reduction.

## 1. Introduction

In recent years, with the rapid development of wireless communication technology and high-frequency devices, microwave-absorbing materials have attracted more and more attention in the military and civilian fields [1,2]. In actual use, designing the wave absorber into a different shape or a coating filled with magnetic metal particles plays a vital role in wave absorption. The most commonly used magnetic metal particles are carbonyl iron particles (CIPs), characterized by high saturation magnetization (*M_s_*), uniform spheres, smaller than 10 microns, a narrow particle size distribution, good microwave absorption, and higher cost-effectiveness applications. However, carbonyl iron is a highly reactive chemical substance, and it is easily oxidized by oxygen in the presence of water [3] or a temperature higher than 200 °C [4]. As a result, as the carbonyl iron particles are gradually oxidized, the electromagnetic performance drastically deteriorates. This shortcoming limits the post-treatment of carbonyl iron powder under high-temperature or complex environments.

To solve this problem, covering materials are used to coat the iron particles, which insulate the interaction of oxygen and iron cores, thereby preventing the oxidation of the carbonyl iron powder. In the past decades, various heat-resisting materials have been used to coat iron powders for improvement of thermal stability, such as aluminum phosphate [5], polyaniline [6], Al_2_O_3_ [7], Al [8,9], Ag [10], Co [11], Ni [12], and silica [13,14,15,16]. These materials improved the thermal stability of the coated samples to varying degrees. However, few works were devoted to studying the variation of the samples’ static magnetic and electromagnetic properties before and after heat treatment, which is very important for practical applications.

In this article, we fabricated a silica/carbonyl iron powder (SiO_2_@Fe) core-shell structure. In order to study the thermal stability and electromagnetic properties after heat treatment, the silica coated and uncoated samples were annealed at different temperatures in an air atmosphere. Then, a series of measurements were performed, such as crystal structure, hysteresis loop, and microwave permeability. In addition, we also studied the relationship between the mass increase ratio after heat treatment and the high-temperature annealing magnetic properties.

## 2. Materials and Methods

Raw CIPs were purchased from Jilin Zhuochuang New Materials Co., Ltd., Jilin, China. The powders were washed in acetone at 50 °C with refluxing for 2 h, followed by drying under vacuum at 50 °C for 3 h; 4.6 g TEOS and 2.8 g deionized water were mixed with 40 mL acetone at room temperature by mechanical stirring for half an hour in a flask, and then 100 g washed iron powders were added to the mixed solution, and the stirring speed increased to 200 rpm/min. After 2 h, 1 mL ammonia solution (25%) was dropped into the flask to promote the progression of the hydrolysis reaction. After 24 h, the product was washed with acetone three times on a suction filter and then dried under vacuum at 50 °C for 6 h. For convenience, the raw CIPs and SiO_2_@Fe powders were named sample A and sample B, respectively

The crystal structures of the samples were analyzed by X-ray diffraction (XRD) on a diffractometer (Philips Panalytical X’pert, Amsterdam, Holland) with Cu Kα radiation. Photos were taken on a field emission scanning electron microscope (Hitachi S-4800, Tokyo, Japan) (SEM) and transmission electron microscope (TEM) (JIM-2010 Hitachi Tokyo Japan). The samples for microwave electromagnetic properties measurements were mixed with paraffin (mass ratio of 15%) and pressed into a ring shape with a 7.00 mm outer diameter and 3.00 mm inner diameter with a thickness of 2 mm. The scattering parameters (S_11_, S_21_) were measured by a network analyzer (Agilent Technologies E8363B, Santa Clara CA, United States) in the range of 1~18 GHz. All measurements were performed at room temperature.

## 3. Results and Discussion

Figure 1a shows the SEM photo of sample A, and the raw carbonyl iron powders were ball-shaped particles with diameters ranging 1~3 μm. Figure 1b shows the TEM image of sample A, and it seems that the particles of sample A had a rough surface. Figure 1c indicates the EDS spectrum of sample A; Fe, C, and O elements. Figure 1e shows the SEM picture of sample B, and the coated particles had a similar shape compared with the raw powders. Figure 1f clearly shows a thin and complete layer on the particle’s surface in sample B with a thickness of around 90 nm. Furthermore, the EDS spectrum of sample B indicates the presence of silicon on the surface of particles after coating treatment, as shown in Figure 1g. Figure 1d,f show the particle size distribution calculated from the SEM photos. It can be seen that the particle size of both samples varied from 0.5 to 5.5 μm.

The XRD patterns of sample A and sample B from 10° to 90° are shown in Figure 2. There were three peaks at 2θ equal to 44.6°, 65.1°, and 82.4° in the XRD pattern of sample A, and the peaks can be indexed to the (110), (200), and (211) planes of cubic α-Fe (#87-0731). The pattern of sample B was similar to that of sample A. There was no trace of SiO_2_ in sample B, suggesting the coating should be an amorphous structure. The FT–IR spectra of sample A and sample B are shown in Appendix A. It was observed that there were three peaks corresponding to the Si-O-Si bands. In summary, all the evidence indicates that the raw carbonyl iron powders were successfully coated with a tight amorphous silica layer. The thermal gravity (TG) curves of sample A and sample B are showed in Appendix A.

Sample A and sample B were annealed at 220 °C, 250 °C, and 300 °C for 6 h in an air atmosphere and then naturally cooled to room temperature. Sample A annealed at different temperatures was sintered into agglomerates, and the color part turned red, indicating that it was severely oxidized. However, no apparent aggregation was observed for all annealed samples B, and there was almost no change in color. Therefore, the mass ratio of samples A and B before and after annealing was calculated.

Figure 2 shows the XRD spectrum of sample A before and after annealing at 300 °C for 6 h in an air atmosphere. It can be observed that there were two independent phases in the annealed sample A; the primary phase was magnetite (PDF#75-0033), and the rest was α-Fe (PDF#87-0721). After annealing, the corresponding increase in the mass ratio of sample A reached 23% (Table 1), which is lower than the theoretical increase in the mass ratio of α-Fe completely converted into magnetite by 38%. The results show that about 60.8% of the iron atoms in the annealed sample A were oxidized, resulting in a significant change in magnetization. According to the FWHM of the diffraction peak, the grain size of sample A increased significantly.

Figure 3 shows the hysteresis loop for sample A before and after annealing. It was found that the *M_s_* of annealed sample A drastically dropped sharply from 201.5 to 112.5 emu/g, and the *H_c_* rose from 7.5 to 180 Oe. The decrease in *M_s_* is mainly due to the presence of magnetite with *M_s_* lower than that of α-Fe in annealed sample A. The size distribution of both samples was 0.5 to 5.5 μm, which is much larger than the single-domain critical size of iron particles [17]. Therefore, carbonyl iron powder’s internal magnetic domain is a multi-domain state. Under the action of an external magnetic field, the magnetization reversal process is dominated by domain wall motion.

According to reports, the *M_s_* of magnetite is about 75 emu/g [18]. Furthermore, the formed magnetite may act as a “pin” preventing the reversal of the magnetization of the whole particle, resulting in an increase in *H_c_*.

Figure 4a shows the real part (*ε*′) and imaginary part (*ε*″) of the permittivity of sample A before and after annealing at 300 °C from 2 to 18 GHz. It was found that the *ε*’ and *ε*″ of sample A had a considerable increase after annealing. Figure 4b indicates the real part (*μ*′) and imaginary part (*μ*″) of the permeability of sample A before and after annealing at 300 °C in the measured range. It was found that the *μ*′ and *μ*″ of sample B both dropped sharply after annealing within the measured range. There were two peaks in the *μ*″ spectrum of sample A: the first peak was around 5.5 GHz and the second around 10 GHz, which can be ascribed to domain wall motion at lower frequencies and spin rotation at higher frequencies [19].

According to the transmit-line theory [20], the reflection loss (RL) of sample A before and after annealing at different thicknesses in the range of 2–18 GHz was calculated by the following equations:(1)$$RL=20\mathrm{lg}\left|\frac{Z_{in}-Z_{0}}{Z_{in}+Z_{0}}\right|\;$$
(2)$$Z_{in}=Z_{0}\sqrt{\frac{\mu_{r}}{\varepsilon_{r}}}tanh\left(j\frac{2\pi f}{c}d\sqrt{\mu_{r}\varepsilon_{r}}\right)$$
where *Z_in_* is the impedance of the incident wave at the interface between the free space and the material, also called the input impedance; *Z*_0_ is the impedance of the incident wave in free space, called the intrinsic impedance; *μ_r_* = *μ*′ − *jμ*″ is the complex permeability, *ε_r_* = *ε*′ − *jε*″ is the complex permittivity; *c* is the speed of light in vacuum; *d* is the thickness of the absorber; and *f* is the frequency.

It is worth noting that in the first equation, RL can reach the minimum value when |*Z_in_*−*Z*_0_| is infinitely close to zero. Therefore, the impedance matching value *Z* can be defined as [21]:(3)$$Z=\left|Z_{in}/Z_{0}\right|$$

The closer *Z* is to 1, the better the impedance matching. Figure 5a,b show the RL map of sample A before and after annealing in the frequency range of 2.0~18.0 GHz with varied absorber thickness from 1.0 to 5.0 mm. It can be observed that the area with qualified microwave absorption (<−10 dB, 90% absorption) in the annealed sample A was significantly suppressed compared to the former. It is widely accepted that the RL intensity and qualified absorption area are the basis for evaluating an eligible absorber [5]. The optimal RL value of sample A was −41.6 dB at 9.7 GHz, and the thickness was 1.875 mm. However, the optimal RL value of the annealed sample A was only −30 dB at 3.8 GHz and the thickness was 3.7 mm. Compared with the RL intensity, the qualified absorption area was more important, because when the RL value was lower than −10 dB, the absorption efficiency was within the acceptable range from 90% to 100% [22]. On the other hand, the bandwidth for RL < −10 dB of sample A was 9.2 GHz, and when the thickness was 1.5 mm, it covered the entire X and Ku bands. However, under the same thickness, the absorption band of the annealed sample A only covered part of the X and Ku bands, and the corresponding bandwidth was reduced to 4.9 GHz. Therefore, it can be confirmed that the microwave absorption capacity of sample A had a considerable decrease after annealing.

After heat treatment, sample A became a composite of magnetite and α-Fe, and the grain size increased. The presence of the magnetite weakened the interaction of the magnetic particles, resulting in a decrease in the values of *μ*′ and *μ*″ after annealing [5]. In addition, a large number of defects may be generated on the new interface between magnetite and α-Fe in the annealed sample A during the annealing process, which acts as a new polarization center, resulting in a sharp increase in *ε*′ and *ε*″ [23]. As a result, the gap between the complex permittivity and the magnetic permeability was larger than the former, resulting in a decrease in the impedance matching value *Z*.

Figure 6a,b show the relationship between Z in sample A and the frequency from 2 GHz to 18 GHz before and after annealing when the absorber thickness was changed from 1 to 5 mm. It was observed that the Z value of the annealed sample A was significantly lower than that of sample B at all thicknesses in the measurement range. The maximum value of Z for sample A was 0.54 at 4.4 GHz, but after annealing, the maximum value of Z decreased to 0.33 at 3.8 GHz. Good microwave absorption can be achieved when Z is close to 1. However, the Z value of the annealed sample A was far from 1, which means that the impedance was not well matched and more microwaves were reflected instead of entering the absorber interface. All of the facts demonstrate the microwave-absorbing ability deteriorated for sample A after annealing, which can be ascribed to the poor impedance matching.

Figure 2 shows the XRD spectra of sample B before and after annealing at 300 °C. It was observed that the spectrum of sample B after annealing maintained the same trend as that of sample B, and there was only one α-Fe phase in the sample. After annealing, the corresponding mass ratio increase in sample B was 3.8%, and the calculated iron atom oxidation ratio was 10% (Table 1), which is much lower than that of the annealed sample A, indicating that oxidation was basically suppressed by the silica coating.

Figure 3 shows the magnetic hysteresis loops of sample B before and after annealing in air atmosphere at 250 and 300 °C. It can be found that the *M_s_* of sample B gradually decreased with the elevated annealing temperature, while *H_c_* increased (insert table in Figure 3). On the other hand, it was noticed that as the annealing temperature increased, the incremental mass ratio of sample B after annealing increased from 0.5% to 3.8% (Table 1), and the calculated ratio of the iron oxide atoms increased from 1.3% to 10.0%. We believe that the incremental mass of sample B after annealing was from the magnetite in it, which was similar to that in the annealed sample A.

Figure 4c shows the *ε*′ and *ε*″ of the annealed sample B from 2 to 18 GHz. After annealing, the *ε*′ and *ε*″ of sample B maintained the same trend in the measurement range compared with that of the un-annealed sample, and the initial value increased with the increase in annealing temperature.

Figure 4d shows the *μ*′ and *μ*″ spectra of the annealed sample B from 2 to 18 GHz. It can be observed that the permeability spectra of sample B annealed at 220 °C and 250 °C were similar to those of sample B, and the value of *μ*′ and *μ*″ slightly decreased when the annealing temperature increased. However, when the annealing temperature increased to 300 °C, both the real and imaginary parts of sample B apparently decreased. This means that the silica protective coating started to lose its effectiveness at this temperature. This could be caused by the degradation of SiO_2_ coatings, which is determined by delamination and crack formation in a high-temperature environment [24]. Further evidence can be seen in Appendix A. In terms of the incremental mass ratio after annealing, 2% was a tipping point for permeability reduction.

Figure 5c–f show the RL maps of sample B before and after annealing at 220 °C, 250 °C, and 300 °C, respectively, with varied absorber thickness from 1 to 5 mm in the frequency range 2~18 GHz. The optimal RL of sample B was −41 dB at 9 GHz for a thickness of 2.0 mm. When the annealing temperature was 220 °C, 250 °C, and 300 °C, the optimal RL of the samples was –39.3 dB at 8.4 GHz for a thickness of 2.0 mm, –42.6 dB at 8.2 GHz for a thickness of 2.0 mm, and −39.6 dB at 10.6 GHz for a thickness of 1.9 mm, respectively. It was also observed that the bandwidth for RL<–10 dB of sample B decreased from 8.8 GHz to 7.2 GHz with increased annealing temperature when the absorber thickness was 1.5 mm. From this point of view, the microwave absorption performance of sample B was not significantly affected after annealing. 

Figure 6c–f indicate the impedance matching value Z of sample B (a) and sample B annealing at 220 °C (b), 250 °C (b), and 300 °C (d) with varied absorber thickness from 1 to 5 mm in the frequency range from 2 to 18 GHz. For all samples, the highest Z value occurred when the thickness was 5 mm. The maximum Z value of sample B was 0.52. For the annealed samples, the maximum values were 0.51, 0.49, and 0.43 when the annealing temperature was 220 °C, 250 °C, and 300 °C, respectively. It was observed that when the annealing temperature increased, the Z value decreased slowly, which means that sample B maintained a good impedance match compared with sample A.

As described above, the gap between the complex permittivity and magnetic permeability of sample A after annealing treatment increased, resulting in poor impedance matching. In addition, we noticed that the changes in the complex permittivity of samples A and B during the heat treatment were greater than the changes in magnetic permeability. It is speculated that dielectric loss may play an important role in microwave absorption. The dielectric loss process can be explained by the Debye theory [25,26,27]; the relationship between *ε*′ and *ε*″ is written as:(4)$${\left(\varepsilon ’-\frac{\left(\varepsilon_{s}+\varepsilon_{\infty }\right)}{2}\right)}^{2}+{\left(\varepsilon^{\prime \prime }\right)}^{2}=\frac{{\left(\varepsilon_{s}+\varepsilon_{\infty }\right)}^{2}}{2}\;\;\;$$
where *ε_s_* is the static permittivity, and *ε*_∞_ is relative permittivity at infinite frequency. The shape of the curve is a semicircle, called to a Cole–Cole semicircle [28,29]. In detail, enhancement of the Debye dipolar relaxation is accompanied by an increased number of semicircles, an expanded semicircle radius, and a higher frequency position. Figure 7 shows the relationship of sample A and B between *ε*′ and *ε*″ before and after annealing. 

It was clearly observed that the Cole–Cole semicircles of the annealed sample A had the largest radius and highest frequency position, indicating the strongest Debye dipolar relaxation. The strongest Debye relaxation not only brings about supreme dielectric loss, but also causes improper impedance matching.

## 4. Conclusions

Silica-coated CIPs with a core-shell structure were fabricated and annealed at elevated temperature in an air atmosphere to investigate the thermal stability and the variation of magnetic properties. It was found that the coated CIP maintained good microwave absorption performance after annealing compared to the uncoated one. It was confirmed that the silica layer significantly prevented the oxidation of the CIP, which leads to poor impedance matching and microwave absorption performance. It is worth mentioning that the mass increase ratio of the sample after annealing was a reference value for studying the change in its magnetic properties. With the increase in the mass ratio after annealing, the *M_s_* decreased and *H_c_* increased, accompanied by the rise in the *ε*′ and *ε*″ and a reduction in the *μ*′ and *μ*″. It is a reliable and straightforward method to measure the oxidation resistance of coated iron powder by calculating the incremental mass ratio of samples before and after annealing. In this article, an incremental mass ratio less than 2% indicated that the electromagnetic properties of the sample had not changed much compared to those before annealing and had a good impedance match. This article provides a new perspective for studying the high-temperature resistance of soft magnetic materials.

## Figures and Tables

**Figure 1 materials-15-02499-f001:**
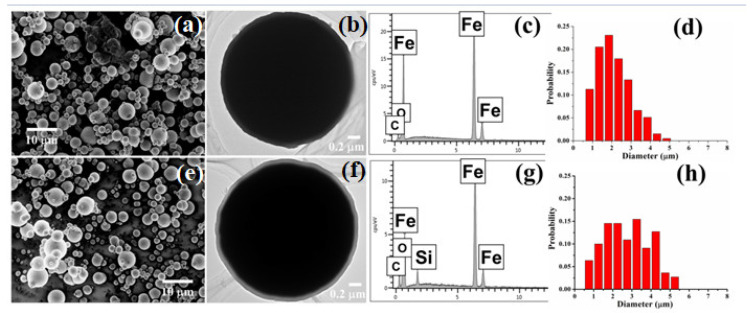
Scanning Electron Microscope (SEM) (**a**) and Transmission Electron Microscope (TEM) (**b**) images of sample A, EDS spectrum of sample A for the selected area (**c**), and particle size distribution of sample A (**d**); SEM (**e**) and TEM (**f**) images of sample B, EDS spectrum of sample B for the selected area (**g**), and particle size distribution of sample B (**h**).

**Figure 2 materials-15-02499-f002:**
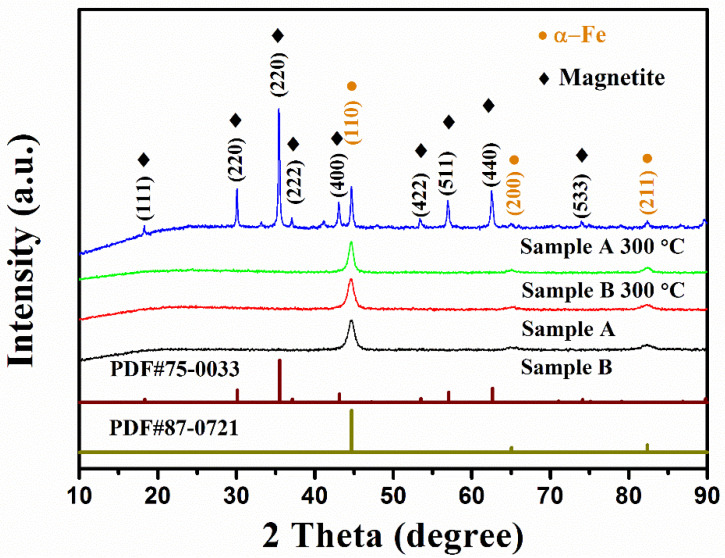
The XRD patterns of samples A and B before and after annealing at 300 °C.

**Figure 3 materials-15-02499-f003:**
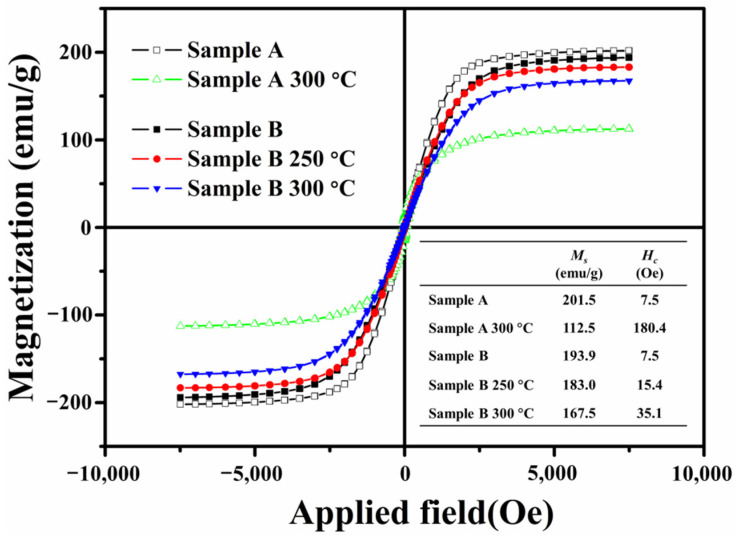
The hysteresis loops of samples A and B before and after annealing at 250 °C and 300 °C.

**Figure 4 materials-15-02499-f004:**
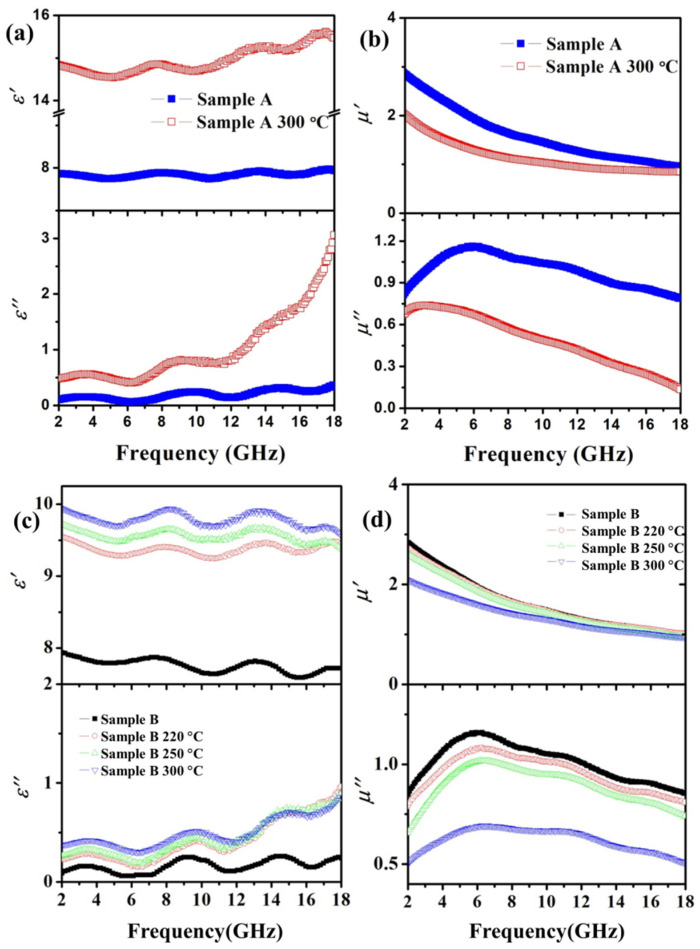
The permittivity spectra (**a**) and permeability spectra (**b**) of sample A before and after annealing at 300 °C; the permittivity spectra (**c**) and permeability spectra (**d**) of sample B before and after annealing at 220 °C, 250 °C, and 300 °C.

**Figure 5 materials-15-02499-f005:**
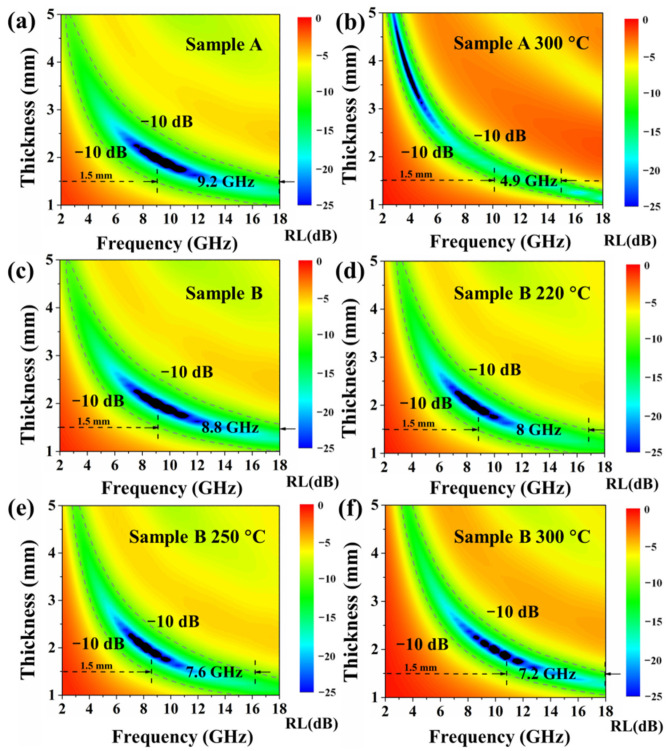
Reflection loss (RL) maps of sample A (**a**), sample A annealing at 300 °C (**b**), sample B (**c**), sample B annealing at 220 °C (**d**), sample B annealing at 250 °C (**e**), and sample B annealing at 300 °C (**f**).

**Figure 6 materials-15-02499-f006:**
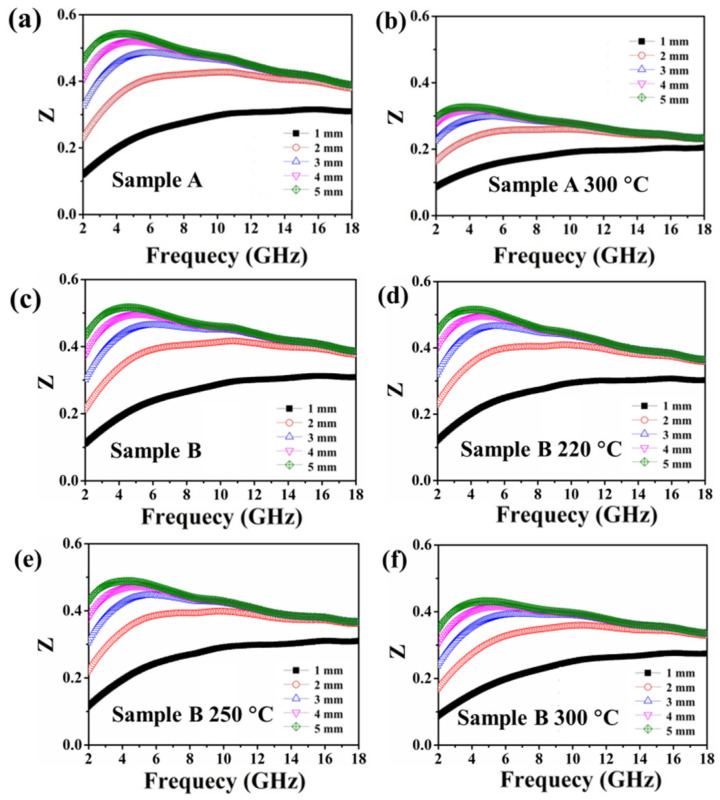
The impedance matching value Z of sample A (**a**), sample A annealing at 300 °C (**b**), sample B (**c**), sample B annealing at 220 °C (**d**), sample B annealing at 250 °C (**e**), and sample B annealing at 300 °C (**f**).

**Figure 7 materials-15-02499-f007:**
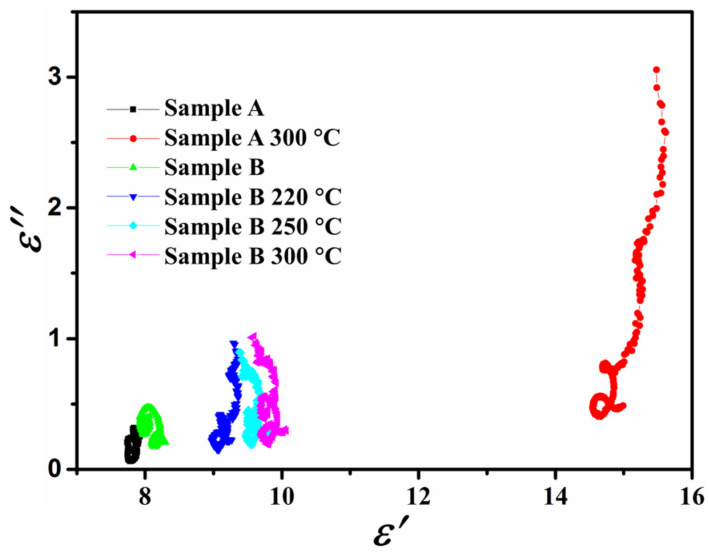
Typical Cole–Cole semicircles of *ε*′ vs. *ε*″ for sample A and B before and after annealing.

**Table 1 materials-15-02499-t001:** Incremental mass ratio and ratio of oxidized Fe atoms for sample A and sample B after annealing at different temperatures.

Annealing Temperature	Sample A Incremental Mass Ratio	Sample A Ratio of Oxidized Fe Atoms	Sample B Incremental Mass Ratio	Sample B Ratio of Oxidized Fe Atoms
220 °C			0.5%	1.3%
250 °C			2.0%	5.0%
300 °C	23%	60.8%	3.8%	10.0%

## Data Availability

Not applicable.

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
