# Peer review of "A Study on the Static Magnetic and Electromagnetic Properties of Silica-Coated Carbonyl Iron Powder after Heat Treatment for Improving Thermal Stability"

_materials, 2022, doi:10.3390/ma15072499_

Round 1
Reviewer 1 Report
-
Regarding the organisation of the manuscript, the correlation between the written results and the images presented is difficult. It would be convenient if the figures describing the same type of characterisation, in different samples, were presented together in order to make the interpretation more fluid. Another point to consider would be the sections of the paper; section 3 presents the results and their discussion, while section 4 would correspond to the conclusion. It would be appropriate if the authors could correct this.
- When the authors describe the experimental procedure, what is the reason for adding ammonia (catalyst) to the mixture of TEOS, water and acetone one hour later?
- In table 1, where the authors show the proportion of oxidised Fe atoms, the increased percentage could be due to sample inhomogeneity or damage to the silica coating? TEM imaging or chemical characterisation (IR, EDS, XPS) after annealing at different temperatures would be adequate to evaluate the coating.
Author Response
- We have redrawn the figures and added a description for each figure. The places mentioned in the text have been modified. The name of section 3 was modified to “Results and their discussion”, the name of section 4 was changed to “Conclusion”.
- The carbonyl iron powder we used in the experiment may have a certain agglomeration after long-term storage. Therefore, strong mechanical stirring is required during the silica coating process to disperse the carbonyl iron powder to a monodispersed state to achieve a uniform coating as possible. In addition, the solution of TEOS, water, and acetone needs a certain time to wet the surface of carbonyl iron powder. Given the above reasons, it is beneficial to form a uniform silica coating layer after adding the carbonyl iron powder to the mixed solution for one hour and then adding ammonia water to catalyze the reaction.
- The TEM photos of Sample B annealing at 300 °C are shown in sFig.3. sFigure. 4 shows the XPS spectra of Sample B, Sample B annealing at 250 °C, and Sample B annealing at 300 °C from 0 to 800 eV. The TEM and XPS measurement clearly show that there are not silica coatings at the surface of Sample B annealing at 300 °C. So, it’s reasonable that the silica coating starts to lose its effectiveness after annealing at 300 degrees so that it falls off the surface of the iron powder.

Reviewer 2 Report
This work present study on the static magnetic and electromagnetic properties of silica coated carbonyl iron powder after heat treatment for improving thermal stability.
However, some critical issues should be raised:
1) The authors should make statistics for particles size distribution from more SEM images The information have to be present in histograms. It is obvious that there are a broad size distribution but it is not discussed in the text.
2) From results presented in this way is not possible to estimate the proper particles size, the domain state which is very important for the behavior of the magnetic particles in magnetic field and their microwave properties.
3) The authors have to show the measured results for RL to compare with the calculated.
Author Response
- Particle size distribution statistics have been added. Figure. 1(d) and (h) show the particle size distribution statistics of sample A and sample B, respectively. The range of particle size distribution is also included in the manuscript. “Fig. 1 (d) and (f) show the particles size distribution calculated from the SEM photos. It can be seen that the particles size of both the samples varied from 0.5 to 5.5 μm” is added at the end of the 6th paragraph.
- The size of carbonyl iron particles is much larger than the single-domain critical size, so it is in a multi-domain state. “It was known that the size distribution of both samples is 0.5 to 5.5 µm, which is much larger than the single-domain critical size of iron particles [17]. Therefore, carbonyl iron powder's internal magnetic domain is a multi-domain state. Under the action of an external magnetic field, the magnetization reversal process is dominated by domain wall motion.” is added at the end of the 10th paragraph.
3. To explain the behavior of the magnetic particles in the microwave field, “There are two peaks in the µ'' spectrum of Sample A, the first peak is around 5.5 GHz, and the second is about 10 GHz which can be ascribed to domain wall motion at lower frequencies and spin rotation at higher frequencies [19].” is added on the end of the 11th paragraph.
Two references have been added.
- Our institute has no devices to test RL using the arch method. We have connected to another lab, and they have equipment and instruments. But there were a lot of samples in the queue, and they told me that they would prioritize people from their institutes to test, and my samples had to be in the back, and it might take a long time to finish the test. So currently, I can't find a lab to complete this test anytime soon.
On the other hand, if the accuracy and thickness of the sample are controllable, the RL measured and calculated by the coaxial cable method is the same as the RL measured by the arch method. Papers supporting this result include J. Mater. Chem. C, 2017,5, 2175 and Journal of Materials Science: Materials in Electronics 2019, 30, 14480. The Doi is:
https://doi.org/10.1039/c6tc05057c
https://doi.org/10.1007/s10854-019-01817-9